# EXAM: Ex-vivo allograft monitoring dashboard for the analysis of hypothermic machine perfusion data in deceased-donor kidney transplantation

**Simon Schwab**[1,2]*, **Hélène Steck**[1], **Isabelle Binet**[3], **Andreas Elmer**[1], **Wolfgang Ender**[3], **Nicola Franscini**[1], **Fadi Haidar**[4], **Christian Kuhn**[3], **Daniel Sidler**[5], **Federico Storni**[6], **Nathalie Krügel**[1], **Franz Immer**[1]

1 Swisstransplant, Bern, Switzerland, 2 Epidemiology, Biostatistics and Prevention Institute (EBPI) & Center for Reproducible Science (CRS), University of Zurich, Zurich, Switzerland, 3 Nephrology and Transplantation Medicine, Kantonsspital St. Gallen, St. Gallen, Switzerland, 4 Division of Nephrology, Department of Medicine, University Hospital of Geneva, Geneva, Switzerland, 5 Department of Nephrology and Hypertension, Inselspital, Bern University Hospital, University of Bern, Bern, Switzerland, 6 Department of Visceral Surgery and Medicine, Inselspital, Bern University Hospital, University of Bern, Bern, Switzerland

* simon.schwab@swisstransplant.org

## Abstract

Deceased-donor kidney allografts are exposed to ischemic injury during ex vivo transport due to the lack of blood oxygen supply. Hypothermic machine perfusion (HMP) effectively reduces the risk of delayed graft function in kidney transplant recipients compared to standard cold storage. However, no free software implementation is available to analyze HMP data for state-of-the-art visualization and quality control. We developed the tool EXAM (ex-vivo allograft monitoring) as an interactive analytics dashboard. We wrote functions in the R programming language to read, process, and analyze HMP data from the LifePort kidney transporter (Organ Recovery Systems, USA). Time series for pressure, flow rate, organ resistance, and temperature are visualized, and relevant statistical indicators have been developed. We explain how data were processed, and indicators were calculated, and we present summary statistics for N = 255 kidney allografts receiving machine perfusion in Switzerland between 2020 and 2023. Median (interdecile range, IDR) of the main indicators were as follows: perfusion duration 5.18 hours (2.29−11.2), flow rate 110 ml/min (52.9−167), ice temperature 1.97˚C (1.53−3.07), and perfusate temperature 6.68˚C (5.58−8.36). We implemented the dashboard to identify issues, such as atypical perfusion parameters, high ice, or high perfusate temperature to inform transplant centers for quality assurance. In conclusion, EXAM is a free tool that statisticians and data scientists can quickly deploy to enable quality control at transplant organizations that use LifePort kidney transporters. An online viewer is available at https://data.swisstransplant.org/exam/.

**Data Availability Statement:** The source code and anonymized example perfusion datasets are available online: https://github.com/Swisstransplant/EXAM. An online version of the tool is available at https://data.swisstransplant.org/exam/.

**Funding:** The author(s) received no specific funding for this work.

**Competing interests:** The authors have declared that no competing interests exist.

## Author summary

In deceased-donor kidney transplantation, a widely used treatment is to perfuse the kidney with a cold preservation solution during transport (hypothermic machine perfusion). High-quality evidence shows that this intervention reduces the risk of delayed graft function in the recipient after transplantation. There are only a few devices available, among them the LifePort kidney transporter, which records the time series of the perfusion and temperature parameters (vascular resistance, flow rate, ice, and perfusate temperature). Currently, no software exists to read, process, visualize, and perform statistics with Life-Port data. We created EXAM (ex-vivo allograft monitoring), a free tool that provides an online state-of-the-art analytics dashboard. Our work will enable transplant organizations to inspect their data, perform statistics and quality checks to help identify potential problems and optimize the intervention to the benefit of kidney recipients.

## Introduction

Hypothermic machine perfusion (HMP) during transport partially preserves deceased-donor kidney allografts from ischemia-related damage due to the lack of blood oxygen. After retrieval, the kidney is connected to the machine and constantly perfused at a hypothermic temperature between 4˚C and 10˚C [1,2]. Commonly used kidney perfusion machines are LifePort (Organ Recovery systems; Itasca, IL, USA), KidneyAssist (XVIVO; Groningen, Netherlands), and Waters RM3 (Rochester, MI, USA). High-level evidence from a Cochrane systematic review suggests a 23% risk reduction for delayed graft function (DGF) after transplantation compared to static cold storage (risk ratio 0.77; 95% CI from 0.67 to 0.90) [3]. Furthermore, one-year graft survival was also superior, as shown in an international randomized controlled trial [4]; more recent studies add further evidence of the efficacy of machine perfusion [5,6].

In Switzerland, 291 deceased-donor kidneys were transplanted in 2023 [7], and 159 were subjected to machine perfusion. Kidneys that are at a higher risk receive HMP treatment. The criteria for kidney HMP are donation after circulatory death (DCD) or donation after brain death (DBD) with either donor age $\geq$ 70 years or a minimum of two out of three donor criteria: arterial hypertension, cerebrovascular cause of death, or last measured serum creatinine > 130 μmol/L; but these criteria are currently being revised. Each of the six transplant centers in Switzerland is equipped with two LifePort machines.

The devices are equipped with several sensors to monitor the progression of the perfusion characteristics. During HMP, the LifePort device records the following parameters every 10 seconds: systolic and diastolic pressure, flow rate, vascular resistance, and ice and perfusate temperature. Data are stored on the device and can be retrieved using a USB memory stick. LifePort's manufacturer provides software that can create a PDF case report, including plots with the time series of the parameters. First, the data visualization is limited: the time series plots are static and do not show data outside a predefined range. Second, there are no summary statistics provided for statistical analysis and quality control. Therefore, this work aimed to provide an advanced statistical analysis tool to read, process, and analyze LifePort HMP data to evaluate the quality of individual machine perfusion interventions given perfusion and temperature parameters.

We created EXAM—ex-vivo allograft monitoring—a tool to inspect perfusion data interactively. It includes state-of-the-art methodology: an analytics dashboard with interactive plots, visual aids, and relevant statistical indicators for an overall assessment of successful HMP,

enabling better quality control, incidence reporting, and novel research. The tool is available in the R programming language and can be easily deployed by data scientists or statisticians working in transplant organizations.

## Materials and methods

### Data collection

Coded data from deceased kidney donors were collected at the six kidney transplant centers in Switzerland (Inselspital, Bern University Hospital; University Hospital of Geneva; Cantonal Hospital St. Gallen; University Hospital Basel; Lausanne University Hospital; University Hospital Zurich). Each center was equipped with two LifePort perfusion machines (Organ Recovery Systems; ORS).

### Ethics approval and consent to participate

The project was submitted to the Ethics Committee of the Canton of Bern, which granted an exemption from requiring ethics approval (KEK Bern; 2023–00557). Legal regulation in Switzerland follows an internationally applied distinction between research subject to approval and quality assurance that is not subject to approval.

For HMP data from deceased-donor kidneys, consent was impossible to obtain as these persons were ventilated and in intensive care with acute circulatory failure or with a diagnosis of brain death. The data were from deceased-donor kidneys transported to and transplanted at the six Swiss transplant centers: Inselspital, Bern University Hospital; University Hospital of Geneva; Cantonal Hospital St. Gallen; University Hospital Basel; Lausanne University Hospital; University Hospital Zurich.

### Development and implementation

EXAM has been fully developed in the R programming language. The source code is available on GitHub (https://github.com/Swisstransplant/EXAM). An online version can be accessed at https://data.swisstransplant.org/exam/. EXAM was licensed under the GNU Affero General Public License (AGPL v3).

EXAM can be used in both a simple and an advanced configuration. The simple configuration is using the online tool to upload the raw data files to assess an individual case. No data is stored permanently except for the two example cases.

The advanced configuration is to install EXAM on a local computer and load all the available data into the dashboard. This has the advantage that a quality manager can access and review all the cases and data sets simultaneously. This procedure is shown in Fig 1 and consists of two steps: first, the data preparation (reading, processing, and the calculation of statistical indicators for all cases), and second, loading and viewing the data with the dashboard. The data preparation reads all individual LifePort data files and saves them into a single.RData file. The data preparation is required every time a new data set is added so that the latest cases are also shown in the dashboard. After data preparation, the dashboard can be loaded to import the.RData file. This two-step approach is simple, fast, and has the advantage that no server or database is required.

In the advanced configuration, the tool can be installed on a local computer by researchers, data scientists, or quality managers. The requirements are an installation of R, RStudio, and Quarto, which are all free software. Importantly, EXAM requires the Swisstransplant R package swt (https://github.com/Swisstransplant/swt), which provides the core functionality (Table 1).

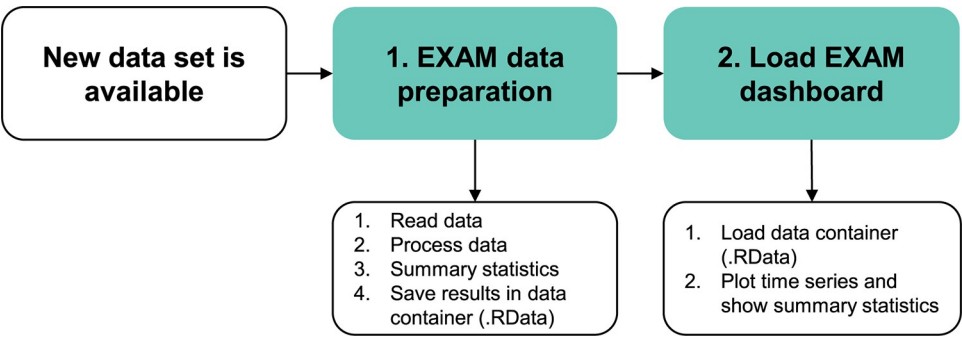

**Fig 1. Schematic overview of EXAM when used in a local computer installation to access all the data available (advanced configuration).**

## Data import

The LifePort raw data is available in two file types. One is a binary raw data file stored directly on the LifePort machine. This file is the best option as no additional software and manual conversion is required. The second type is a plain text ASCII file converted from the binary raw data using the manufacturer's software. Note that both file types contain the same information and a temporal resolution of 1/10 Hz. In other words, they are equivalent from a data analysis perspective.

Most of the data we collected was in the original binary raw data format; however, we sometimes receive ASCII plain text files. We implemented a function lifeport_read() as part of the swt package that can handle both file types. For this aim, the structure of the binary raw data file had to be recovered, i.e., what parts of the binary code contained what information (time series, organ ID, kidney side, serial number, unit ID, start time, etc.). With the swt package, it is now possible to interact directly with LifePort HMP raw data from within R/RStudio.

**Table 1. Summary of main functions and procedures for processing LifePort HMP data. The data is processed in three steps: data import, data processing, and calculation of the statistical indicators.**

| Source | Implementation | Responsibility |
|---|---|---|
| R package swt | Data import swt::lifeport_read() | Reads binary or ASCII raw data files obtained from LifePort machines. A list of three objects is returned containing device information (serial number, unit ID, start time, run time, etc.), organ information (organ ID, kidney side, blood type, etc.), and time series data (systolic pressure, diastolic pressure, flow rate, vascular resistance, ice temperature, and perfusate temperature). |
| | Data processing swt::lifeport_process() | Adds actual clock time given the device start time. Pressure, flow, and vascular resistance time series are smoothed using a moving average with a window size of 15 samples (150 seconds). Temperature time series are not smoothed. |
| | Statistical indicators swt::lifeport_sumstats() | Calculates various statistical indicators based on pressure, flow rate, vascular resistance, and temperature time series data. |
| EXAM | Data preparation Any custom R script or Quarto document | Data preparation pipeline that applies the above swt functions sequentially (simple for loop) to import, process, and calculate statistical indicators for every dataset. The aggregated data is stored in a single RData container. |
| | Quarto dashboard EXAM-app.qmd | Loads the data container and visualizes all the cases in an interactive analytics dashboard based on Quarto dashboard, Plotly, and Shiny. |

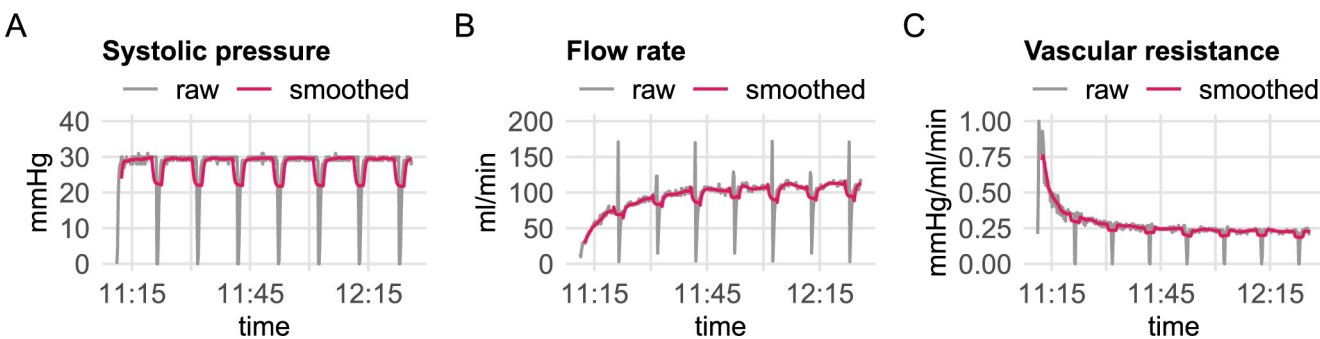

**Fig 2.** Time series for **A** systolic pressure (diastolic pressure not shown), **B** flow rate, and **C** vascular resistance were smoothed (pink line) using a rolling average with a window size of 150 seconds (15 samples).

## Data processing

Data processing with lifeport_process() handles the actual time of the machine perfusion so that the time series in the plots do not start from time zero but at the actual clock time of the start of the HMP; this may be helpful for quality control and incidence reporting to see events against the actual clock time. Furthermore, the time series of pressure, flow rate, and vascular resistance were smoothed (Fig 2); for temperature data, this step was not necessary.

## Statistical indicators

During visual inspection of many different HMP cases, we developed statistical indicators that seem relevant for quality control. The function lifeport_sumstats() calculates these; they are explained in Table 2.

Some conditions must be met to calculate the indications to reflect an accurate summary. For example, when the machine stops the perfusion data is still recorded and the pressure will be 0. Thus, we only calculate the mean pressure across the period where pressure was larger than 0, i.e., the period the kidney was perfused. The same principle affects the mean flow; however, here, we use a cutoff of 5 as flow is never precisely 0, and there is always some residual signal. Another example is the mean perfusate temperature, which is only calculated for time segments with flow larger than 25. Without flow, the perfusate temperature sensor will record a high temperature, but perfusate temperature can be disregarded when the kidney is not perfused.

**Table 2. Statistical indicators for quality control.**

| Class | Indicator | Meaning | Unit | Condition |
|---|---|---|---|---|
| Perfusion indicators | $md_{sys}$ | median systolic pressure | mmHg | pressure must be $> 0$. |
| | $m_{dia}$ | mean diastolic pressure | mmHg | pressure must be $> 0$. |
| | $m_{flow}$ | mean flow rate | ml/min | first 30 min. excluded; flow must be $> 5$. |
| | $m_{res}$ | mean vascular resistance | mmHg/ml/min | first 30 min. excluded; resistance must be $> 0$. |
| | $SD_{res}$ | std. dev of vascular resistance | mmHg/ml/min | first 30 min. excluded; resistance must be $> 0$. |
| Temperature indicators | $m_{ice}$ | mean ice temperature | ˚C | – |
| | $SD_{ice}$ | std. dev. of ice temperature | ˚C | – |
| | $\Delta t_{y>2.5}$ | time duration ice above 2.5˚C | MM:HH:SS | – |
| | $y_{start}$ | perfusate temperature at start | ˚C | window of 5 min. after discarding first 2 min; flow must be $> 5$. |
| | $m_{perf}$ | mean perfusate temperature | ˚C | first 2 min. are excluded and flow must be $> 5$. |
| | $SD_{perf}$ | std. dev. of perfusate temperature | ˚C | first 2 min. are excluded and flow must be $> 25$. |
| | $\Delta t_{y>10}$ | time duration perfusate temperature above 10˚C | MM:HH:SS | first 2 min. are excluded and flow must be $> 25$. |

For the three indicators based on flow rate and vascular resistance, the first 30 min. are excluded as this is the period where the flow rate builds up (and the resistance decreases) until the kidney reaches a steady state. Thus, these indicators are not calculated if the overall perfusion duration is below 30 min.

## Dashboard, data visualization, and reporting

The dashboard has been implemented with Quarto, Shiny, and Plotly [8,9]; see Fig 3. This allows the user to interact with the data, for example, zooming in and out, data value display on hoover, panning, autoscale, reset axes, etc. This highly facilitates data inspection and quality control.

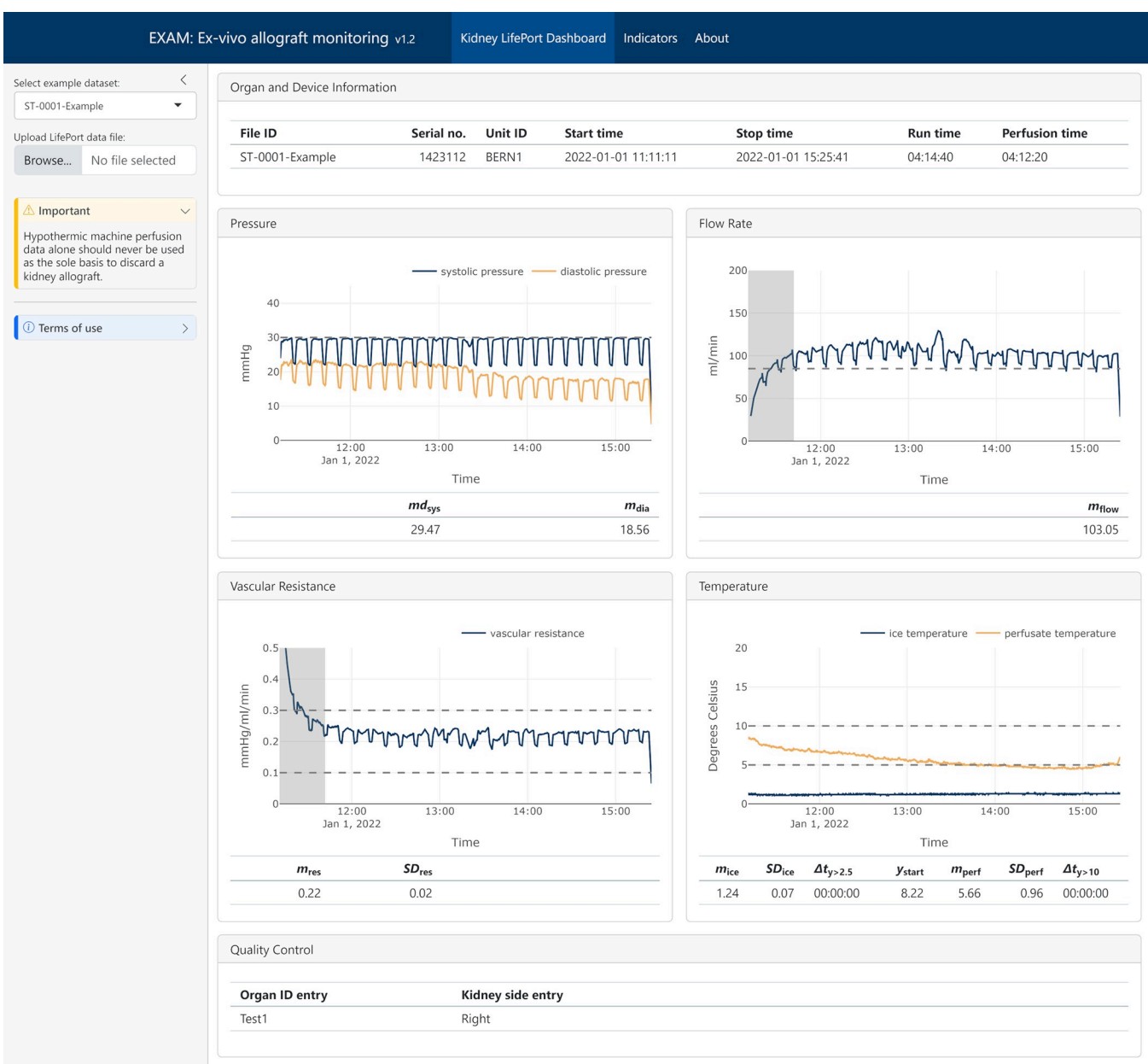

**Fig 3. EXAM dashboard.** Organ and device information is on the top; below interactive plots with time series data and statistical indicators. On the left is a dropdown menu to select cases and a button to upload a LifePort data file.

Visual aids were added to the plots that facilitate interpretation, such as horizontal dashed lines for optimal ranges or critical thresholds, for example, 10˚C as the upper limit of HMP or 5˚C for ice temperature, which is the threshold to produce an alarm by the perfusion machine.

### Validation

We validated the data displayed by the EXAM dashboard (time series) by comparing our figures with the case reports created by the manufacturer's software (ORS Data Station). However, we directly read the time series data from the raw data files with minimal processing (smoothing). The indicators are based on medians, means, or standard deviations and do not require validation.

## Results

### Descriptive statistics

We collected 266 LifePort datasets from 1. January 2020 to 31. December 2023. Eleven cases (4.1%) were excluded due to missing data for calculating the statistical indicators. These were caused by a perfusion time of below 30 minutes, and in such cases, flow and vascular resistance indicators were not calculated. The analysis data set had a sample size of N = 255 kidneys with machine perfusion. These kidneys were from 184 deceased donors, of which 131 (71.2%) were organ donations after circulatory death (DCD), and 53 (28.8%) were donations after brain death (DBD). The median age of the donors was 58 years (range 11–86 years). Summary statistics of the HMP indicators are shown in Table 3; the distributions are shown in Fig 4.

The capability of EXAM and the developed statistical indicators is now highlighted in a short example.

### Example: Comparing perfusate and ice temperature

We compare the temperature profiles of two cases. In Fig 5A, the start perfusate temperature was 8.2˚C, and the mean was 5.7˚C. In Fig 5B, the initial state of the perfusate was 14.9˚C, and the mean was 9.3˚C. The perfusate temperature was above 10˚C for 1 hour and 4 minutes. The

**Table 3. Descriptive statistics (median and interdecile range) for the perfusion duration and the statistical indicators from N = 255 kidney allografts receiving HMP.**

|  | Median across kidneys (N = 255) | Interdecile range (10th–90th percentile) |
|---|---|---|
| Perfusion duration (hours) | 5.18 | 2.29–11.2 |
| Perfusion indicators |  |  |
| Median systolic pressure (mmHg) | 29.4 | 21.7–29.6 |
| Mean diastolic pressure (mmHg) | 18.9 | 15.1–22.4 |
| Mean flow rate (ml/min) | 110 | 52.9–167 |
| Mean vascular resistance (mmHg/ml/min) | 0.21 | 0.11–0.43 |
| SD vascular resistance (mmHg/ml/min) | 0.02 | 0.01–0.09 |
| Temperature indicators |  |  |
| Mean ice temperature (˚C) | 1.97 | 1.53–3.07 |
| SD ice temperature (˚C) | 0.20 | 0.09–0.75 |
| Duration ice temperature > 2.5˚C (min) | 0.83 | 0–379 |
| Mean perfusate temperature (˚C) | 6.68 | 5.58–8.36 |
| SD perfusate temperature (˚C) | 0.94 | 0.46–1.49 |
| Duration perfusate temperature > 10˚C (min) | 0.17 | 0–30.8 |
| Start perfusate temperature (˚C) | 9.15 | 7.16–12.0 |

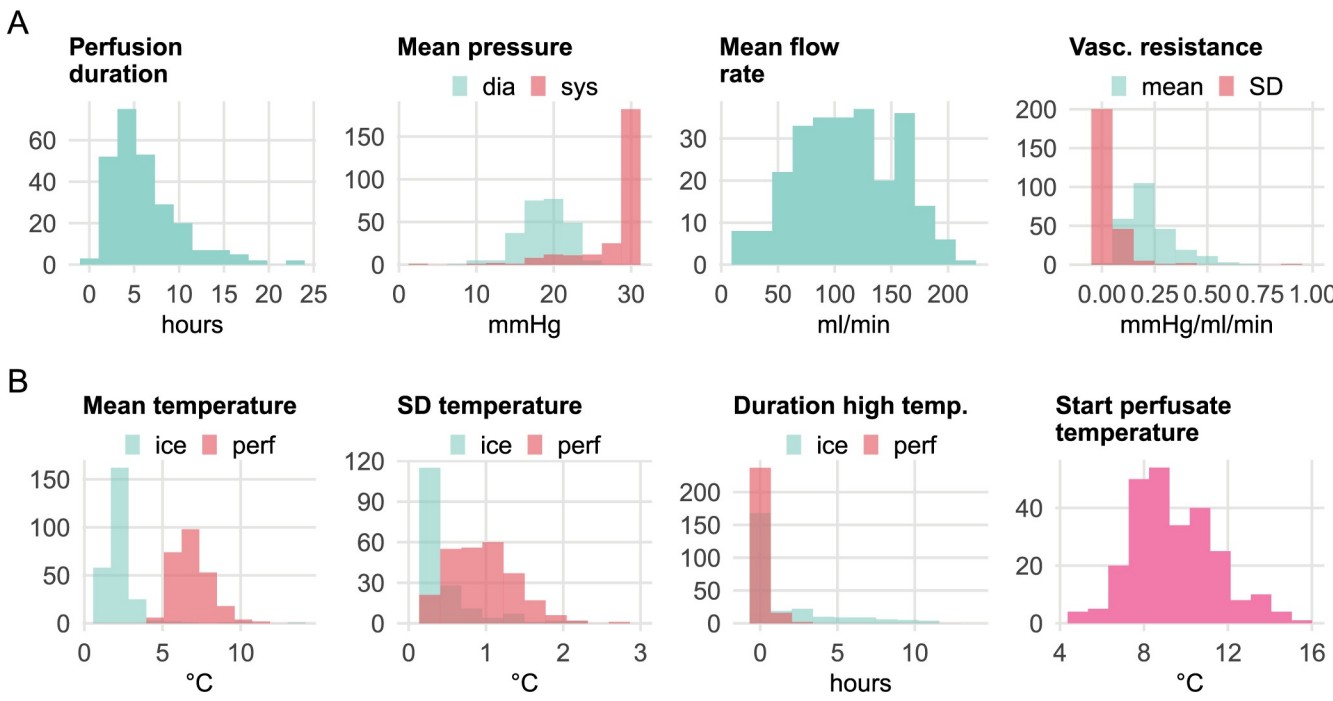

**Fig 4.** Data distributions of **A** perfusion duration and perfusion indicators, and **B** temperature indicators.

difference is also highlighted by an almost 3-fold increase in the variation (0.96°C versus 2.8°C). A smaller variation often reflects a more horizontal, stable curve, while a larger variation indicates a more steeply falling (or rising) curve or any other type of fluctuation. Regarding the ice temperature, both means were below average (1.2°C and 1.8°C) and stable (a slight variation of 0.07°C and 0.11°C across time).

## Discussion

EXAM is a free tool in the R programming language (with Quarto dashboard and R Shiny) for the visual inspection and quality control of HMP data from the LifePort kidney transporter. Its main features are an analytics dashboard with interactive plots and calculation of statistical indicators for quality control. Furthermore, we provide an example for interpreting the statistical indicators concerning temperature.

There is ample evidence that HMP effectively lowers DGF and prolongs graft survival [3,4]. However, this requires optimal conditions in terms of perfusion and temperature during the treatment of the kidney allograft. One criticism of clinical trials is that they are artificial and do not provide real-life evidence. Therefore, quality assurance of the intervention in clinical practice is crucial to achieving the treatment benefit.

EXAM was developed for retrospective analysis of HMP data for quality control purposes. It helps to identify problems and can improve and optimize kidney HMP. Kidney allografts, mainly marginal kidneys, should receive the best treatment in the critical period between procurement and transplantation. This may enhance patient outcomes and help reduce additional costs, for example, when fewer kidney recipients experience DGF after transplantation. EXAM is not a tool for decision-making at the time of transplantation; e.g., HMP data alone should never be used to discard a kidney allograft. Further research may investigate the potential role of anomalies, such as those observed in vascular resistance, in clinical decision-making [10].

A

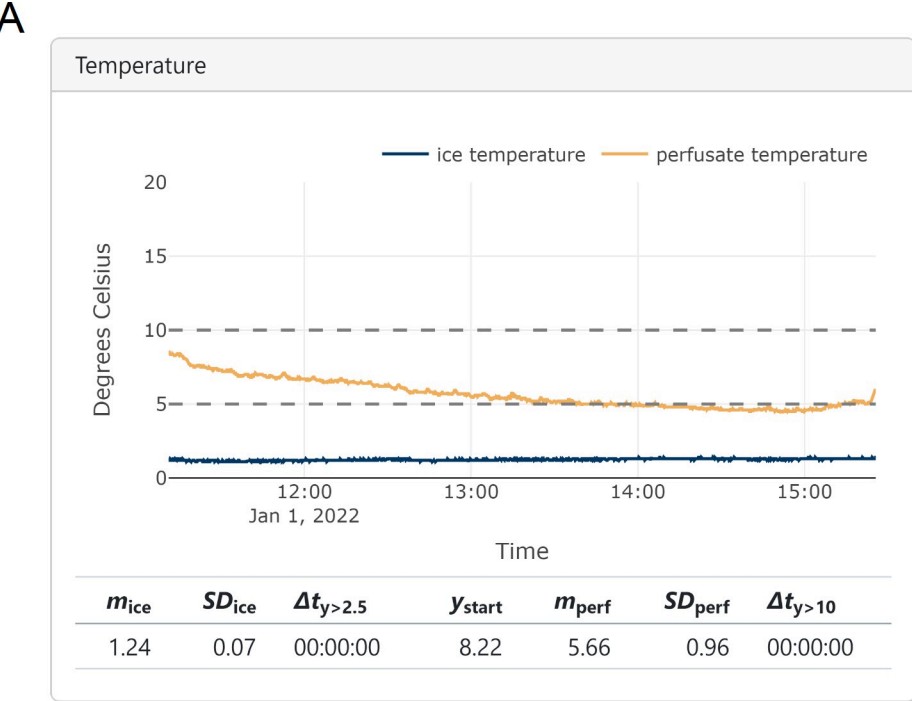

| $m_{ice}$ | $SD_{ice}$ | $\Delta t_{y>2.5}$ | $y_{start}$ | $m_{perf}$ | $SD_{perf}$ | $\Delta t_{y>10}$ |
|---|---|---|---|---|---|---|
| 1.24 | 0.07 | 00:00:00 | 8.22 | 5.66 | 0.96 | 00:00:00 |

B

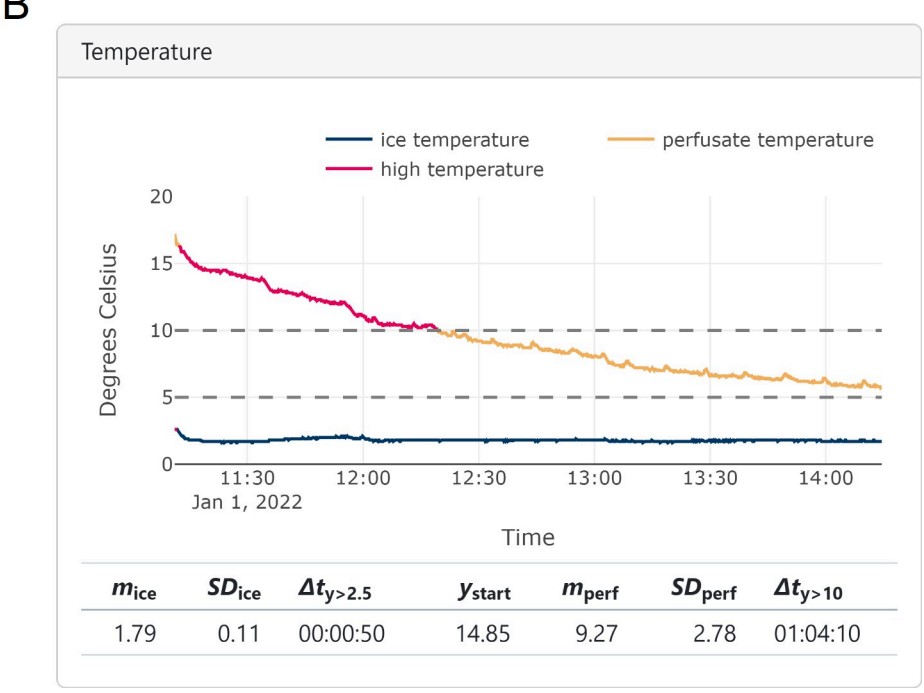

| $m_{ice}$ | $SD_{ice}$ | $\Delta t_{y>2.5}$ | $y_{start}$ | $m_{perf}$ | $SD_{perf}$ | $\Delta t_{y>10}$ |
|---|---|---|---|---|---|---|
| 1.79 | 0.11 | 00:00:50 | 14.85 | 9.27 | 2.78 | 01:04:10 |

**Fig 5.** Two HMP temperature plots with **A** average and **B** elevated perfusate temperature. Visual aids help interpret the data: horizontal lines at 10˚C (the upper limit for hypothermic perfusion) and 5˚C (LifePort device will start an alarm when ice temperature is above 5˚C). The pink line highlights the perfusate temperature above the hypothermic threshold of 10˚C. The indicators are shown below the plots.

A limitation of the study may be the risk in case the manufacturer ORS changes the structure of the data file. Then, EXAM would not work anymore with new data. However, the risk is largely mitigated as the tool is under an open-source license. This would enable everyone, not only Swisstransplant, to modify and share a new version of the tool as long as it is released under the same conditions, i.e., everyone can again modify and share the code, as stated in the AGPLv3 license.

In the future, the dashboard could be expanded to include additional information on error events, such as exceeding the maximum ice temperature, not reaching the required pressure, or check filter. Another idea is to use the Mahalanobis distance to calculate a quality measure based on the various perfusion and temperature indicators. This would identify cases with large distances from the norm (across the indicators' multiple dimensions) and monitor quality over a longer period of time.

Most importantly, the open development of EXAM on GitHub allows data scientists and statisticians to get involved and transplant organizations to use the tool free of charge.

## Conclusions

EXAM is a dashboard based on R and Quarto to visualize HMP data from the LifePort kidney transporter. Quality control with EXAM can identify problems during HMP and support guidelines, checklists, and training for healthcare providers. This assures optimal therapy, which has been demonstrated to be effective in preventing DGF and dialysis after transplantation.

## Author Contributions

**Conceptualization:** Simon Schwab, Hélène Steck, Andreas Elmer, Nicola Franscini, Nathalie Krügel, Franz Immer.

**Data curation:** Hélène Steck.

**Formal analysis:** Simon Schwab.

**Investigation:** Isabelle Binet, Wolfgang Ender, Fadi Haidar, Christian Kuhn, Daniel Sidler, Federico Storni.

**Methodology:** Simon Schwab.

**Project administration:** Simon Schwab, Hélène Steck.

**Software:** Simon Schwab.

**Supervision:** Andreas Elmer, Franz Immer.

**Validation:** Simon Schwab, Hélène Steck.

**Visualization:** Simon Schwab.

**Writing – original draft:** Simon Schwab.

**Writing – review & editing:** Simon Schwab, Hélène Steck, Isabelle Binet, Andreas Elmer, Wolfgang Ender, Nicola Franscini, Fadi Haidar, Christian Kuhn, Daniel Sidler, Federico Storni, Nathalie Krügel, Franz Immer.

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
