## [Decision Letter · Decision Letter 0]

17 May 2024

PDIG-D-24-00069

EXAM: Ex vivo allograft monitoring dashboard for the analysis of hypothermic machine perfusion data in deceased-donor kidney transplantation

PLOS Digital Health

Dear Dr. Schwab,

Thank you for submitting your manuscript to PLOS Digital Health. After careful consideration, we feel that it has merit but does not fully meet PLOS Digital Health's publication criteria as it currently stands. Therefore, we invite you to submit a revised version of the manuscript that addresses the points raised during the review process.

Please submit your revised manuscript within 60 days Jul 16 2024 11:59PM. If you will need more time than this to complete your revisions, please reply to this message or contact the journal office at digitalhealth@plos.org. Please include the following items when submitting your revised manuscript:

We look forward to receiving your revised manuscript.

Kind regards,

Miguel Ángel Armengol de la Hoz, Ph.D.

Section Editor

PLOS Digital Health

Journal Requirements:

1. Please update your online Competing Interests statement. If you have no competing interests to declare, please state: “The authors have declared that no competing interests exist.”

2. Please provide separate figure files in .tif or .eps format only and remove any figures embedded in your manuscript file. Please also ensure that all files are under our size limit of 10MB. You may leave the figure captions or legends in the manuscript.

For more information about how to convert your figure files please see our guidelines: https://journals.plos.org/digitalhealth/s/figures

3. Some material included in your submission may be copyrighted. According to PLOS’s copyright policy, authors who use figures or other material (e.g., graphics, clipart, maps) from another author or copyright holder must demonstrate or obtain permission to publish this material under the Creative Commons Attribution 4.0 International (CC BY 4.0) License used by PLOS journals. Please closely review the details of PLOS’s copyright requirements here: PLOS Licenses and Copyright. If you need to request permissions from a copyright holder, you may use PLOS's Copyright Content Permission form.

Potential Copyright Issues:

Your manuscript (header) and Figure 3 contain branding/a logo. We are not permitted to publish this under our CC-BY 4.0 license, even with permission. We ask that you please remove or replace it.

Additional Editor Comments (if provided):

Reviewers' comments:

Reviewer's Responses to Questions

**Comments to the Author**

1. Does this manuscript meet PLOS Digital Health’s publication criteria? Is the manuscript technically sound, and do the data support the conclusions? The manuscript must describe methodologically and ethically rigorous research with conclusions that are appropriately drawn based on the data presented.

Reviewer #1: No

Reviewer #2: Partly

Reviewer #3: Partly

2. Has the statistical analysis been performed appropriately and rigorously?

Reviewer #1: No

Reviewer #2: No

Reviewer #3: No

3. Have the authors made all data underlying the findings in their manuscript fully available (please refer to the Data Availability Statement at the start of the manuscript PDF file)?

Reviewer #1: No

Reviewer #2: No

Reviewer #3: No

4. Is the manuscript presented in an intelligible fashion and written in standard English?

Reviewer #1: No

Reviewer #2: Yes

Reviewer #3: No

5. Review Comments to the Author

Reviewer #1: 1. Write a separate section for literature review/ Related work.

2. Research requires mathematical & statistical analysis. But no 

 mathematical equations found in the study.

3. Draw a work flow diagram to implement the methodology.

4. Write an algorithm to support the methodology.

5.Need a comparative study with results with this study and related similar types 

 already published articles.(In tabular form)

6. Need few more references from recent publications.

Reviewer #2: The topic of the paper is interesting but it needs major improvements as below:

The abstracts should be rewritten to show more results and what was achieved from the created dashboard. The present abstract is very brief and only highlights the background and methods with minimal findings. 

Remove R programming language from the keywords.

In the method section, it is not clear whether the data collected were anonymous. Was there a framework for data collection?

What was done about missing data?

How were the findings validated?

The parameters collected seem to be incomprehensive. For example patients' clinical characteristics such as comorbidities, risk factors etc... should be included. 

In the results, it is not clear how these indicators helped the patients.

The discussion is rather a conclusion that is under-referenced. The discussion section is too brief and does not justify the findings in terms of the literature. It should be rewritten to highlight the implications of the findings on the literature and on practice.

The conclusion should be more summative and informative

Reviewer #3: Abstract and Author Summary

What is not clear from the abstract and author summary is whether EXAM is a new tool that the authors have developed. The authors say there is currently no free software. Then immediately talk about R which is an open source software. Please clarify this.

Introduction

There same issue with abstract and author summary. EXAM is being introduced without clear information whether this is an existing platform, is it an R package? Did the authors develop it from scratch? Please clarify.

There is a hanging statement in paragraph three… There are two significant ??

Development and Implementation

Based on the links – This looks like a Shiny Dashboard. Is this what the authors call EXAM? If so, then EXAM is not a newly created software, it is a visualization platform which reads data from the LifePort machine. 

If the above paragraph is correct, then the innovation here is the use of shiny dashboards to visualize the data that has traditionally not been visualized that way. 

Then, it is not correct to say there is no free software because what the authors have done is to utilize an existing software (R), which is free, to visualize data.

Data import

Write ASCII in full. Ideally, an acronym needs to be written in full if using it for the first time in the document.

Statistical indicators

I think it is not correct to say statistical indicators. In other words, are there are other indicators that are not “statistical”? I would suggest, just have the topic to read “Indicators” or “Indicators of interest” or “Outcomes”

Results

Table 3: The title talks about median, but the results includes some indicators summarized using mean. What is unclear is, for the indicators summarized using mean, why do we use the 10-90th percentile range? Mean is usually reported alongside standard deviation. Please clarify what is happening here.

Discussion

This is too brief. I find the discussion to be lacking in depth and width. It is a major shortcoming of this manuscript.

I expected the discussion to focus on;

a) Atleat a paragraph on summary of the key findings

b) A few paragraphs to discuss what new knowledge or evidence your research has added in the context of other findings in literature. I don’t see any citations in the discussion. I know of papers that have been published that have used RShiny. This is very important. This should be a major part of the discussion.

c) A paragraph to discuss any strengths for your research

d) A paragraph on the limitations

e) Conclusions

Conclusion:

What is DGF in full? I may have missed it, but as I noted earlier, any acronymn should be written in full if appearing for the first time.

I think the conclusion should focus on what is new, and how can it be scaled up? What policy implications can it have?

6. PLOS authors have the option to publish the peer review history of their article (what does this mean?). If published, this will include your full peer review and any attached files.

**Do you want your identity to be public for this peer review?** For information about this choice, including consent withdrawal, please see our Privacy Policy.

Reviewer #1: Yes: Dr. Sumanta Kuila

Reviewer #2: No

Reviewer #3: No

---

## [Decision Letter · Decision Letter 1]

7 Nov 2024

EXAM: Ex vivo allograft monitoring dashboard for the analysis of hypothermic machine perfusion data in deceased-donor kidney transplantation

PDIG-D-24-00069R1

Dear Dr Schwab,

We are pleased to inform you that your manuscript 'EXAM: Ex vivo allograft monitoring dashboard for the analysis of hypothermic machine perfusion data in deceased-donor kidney transplantation' has been provisionally accepted for publication in PLOS Digital Health.

Best regards,

Miguel Ángel Armengol de la Hoz, Ph.D.

Section Editor

PLOS Digital Health

**Additional Editor Comments (if provided):**

**Reviewer Comments (if any, and for reference):**

Reviewer's Responses to Questions

**Comments to the Author**

1. If the authors have adequately addressed your comments raised in a previous round of review and you feel that this manuscript is now acceptable for publication, you may indicate that here to bypass the “Comments to the Author” section, enter your conflict of interest statement in the “Confidential to Editor” section, and submit your "Accept" recommendation.

Reviewer #2: All comments have been addressed

Reviewer #3: All comments have been addressed

2. Does this manuscript meet PLOS Digital Health’s publication criteria? Is the manuscript technically sound, and do the data support the conclusions? The manuscript must describe methodologically and ethically rigorous research with conclusions that are appropriately drawn based on the data presented.

Reviewer #2: Yes

Reviewer #3: Yes

3. Has the statistical analysis been performed appropriately and rigorously?

Reviewer #2: N/A

Reviewer #3: Yes

4. Have the authors made all data underlying the findings in their manuscript fully available (please refer to the Data Availability Statement at the start of the manuscript PDF file)?

Reviewer #2: Yes

Reviewer #3: Yes

5. Is the manuscript presented in an intelligible fashion and written in standard English?

Reviewer #2: Yes

Reviewer #3: Yes

6. Review Comments to the Author

Reviewer #2: No further comments

Reviewer #3: I am happy with the edits the authors have made. However, I disagree with their response on the discussion. If there is "no new knowledge" (using the author's words), then why publish? Scientific publications are meant to extend knowledge, add new knowledge, filling a knowledge gap.

I still feel the authors need to discuss what their innovation does and what implications it may have both for practice and future research. This is so important to contextualize the paper in the broader body of knowledge.

I still feel the discussion is lacking in depth. This needs to be improved.

7. PLOS authors have the option to publish the peer review history of their article (what does this mean?). If published, this will include your full peer review and any attached files.

**Do you want your identity to be public for this peer review?** For information about this choice, including consent withdrawal, please see our Privacy Policy.

Reviewer #2: No

Reviewer #3: No
